# Neonatal COVID-19 Pneumonia: Report of the First Case in a Preterm Neonate in Mayotte, an Overseas Department of France

**DOI:** 10.3390/children7080087

**Published:** 2020-08-03

**Authors:** Soumeth Abasse, Laila Essabar, Tereza Costin, Voninavoko Mahisatra, Mohamed Kaci, Axelle Braconnier, Roger Serhal, Louis Collet, Abdallah Fayssoil

**Affiliations:** 1Mayotte Hospital, Mayotte Island, 97600 Mamoudzou, France; l.essabar@chmayotte.fr (L.E.); t.costin@chmayotte.fr (T.C.); v.mahisatra@chmayotte.fr (V.M.); m.kaci@chmayotte.fr (M.K.); a.braconnier@chmayotte.fr (A.B.); r.serhal@chmayotte.fr (R.S.); l.collet@chmayotte.fr (L.C.); 2Raymond Poincaré Hospital, APHP, 92380 Garches, France; fayssoil2000@yahoo.fr

**Keywords:** COVID-19, preterm neonate, pneumonia, vertical transmission, length of stay

## Abstract

We report the first case of COVID-19 pneumonia in a preterm neonate in Mayotte, an overseas department of France. The newborn developed an acute respiratory distress by 14 days of life with bilateral ground glass opacities on a chest CT scan and a 6-week-long stay in the neonatal intensive care unit (NICU). This case report emphasizes the need for a cautious and close follow-up period for asymptomatic neonates born to mothers with COVID-19 infection. Vertical transmission cannot be excluded in this case.

## 1. Introduction

On 31 December 2019, the China Center for Disease Control and Prevention (CDC) reported an outbreak of pneumonia of unknown causes in the city of Wuhan [1]. 

This pneumonia was linked to a novel coronavirus, the severe acute respiratory syndrome coronavirus-2 (SARS-CoV-2), named coronavirus disease 2019 (COVID-19) [2]. This disease (COVID-19) spread rapidly to many countries all over the world in a few months, and we are currently still in the midst of a pandemic. The first cases detected in Europe were reported from France on 24 January 2020. 

In Mayotte, one of the French overseas departments in the Indian Ocean with a population estimated at 256,000 and a significant birth rate of 10,000 births per year, a COVID-19 infection was first reported on 14 March 2020. As of 24 June 2020, the French public health agency (Santé Publique France) has identified 2467 cases of COVID-19 on the island [3]. Little is known about its clinical characteristics and outcomes in neonates, especially preterm infants. We report, in the present paper, the first case in Mayotte of neonatal COVID-19 thus far, in a preterm infant who showed clinical symptoms and (presumably) vertical transmission. This case highlights the need for a cautious extended follow-up as well as clear and evidence-based guidelines.

## 2. Case Report

A 36-year-old multiparous woman, with a history of idiopathic bronchiectasis and gestational diabetes, was admitted to the delivery emergencies unit for active labor at 33 weeks of gestation. On admission, her body temperature was 37.4 °C, and her respiratory and hemodynamic parameters were within the normal ranges. Maternal antibiotic therapy and intramuscular corticosteroids for fetal pulmonary maturation were not administrated due to a rapid labor that lead to the vaginal birth, with meconium-stained amniotic fluid, of a preterm male infant. During the vaginal delivery, the mother developed a persistent cough and dyspnea, so she was suspected of being infected with COVID-19, and droplet and contact precautions were immediately initiated. RT-PCR of a nasopharyngeal swab from the mother was performed just after delivery and was positive for SARS-COV2 (130,000 copies/uL) and her thoracic CT scan showed peripheral ground glass opacities associated with unilateral bronchiectasis. The neonate was born weighing 1830 g with an Apgar score of five at 1 min, seven at 5 min, and nine at 10 min. Skin-to-skin contact with his mother was not performed; he was transferred to the neonatal intensive care unit (NICU) on non-invasive intermittent positive pressure ventilation. He was in a closed incubator throughout his admission, and the mother’s access was not permitted in the NICU as she was symptomatic.

On admission, he was clinically stable on non-invasive intermittent positive pressure ventilation with a mild transient respiratory distress that resolved within 24 h; his fractional concentrations of oxygen in inspired air (FiO2) remained at 21%. Furthermore, he was exclusively formula fed and was given prophylactic intravenous antibiotic therapy (Cefotaxim and Gentallin) that was stopped at 48 h as the blood culture was negative and the C-reactive protein (CRP) was normal. A nasopharyngeal swab RT-PCR was performed at 24 h of life and showed a positive result for SARS-COV2 with a viral load of 918,000 copies/uL. The RNA extraction was done with the QIAsymphony DSP Virus/pathogen Midi Kit (QIAGEN®). 

The chest X-ray showed normal lung aeration without pneumonia (Figure 1a). 

The infant remained stable and was switched to nasal continuous positive airway pressure at 2 days of life, and gradually weaned to low nasal flow cannula for 48 h; he was then completely weaned to room air at around 7 days of life.

By 14 days of life, the infant developed a fever with progressive signs of increased breathing difficulties, such tachypnea and chest retractions, but his hemodynamic status remained normal.

A thoracic CT scan revealed bilateral ground glass opacities and consolidations (Figure 1b). Echocardiography showed a mild pericardial effusion (3 mm). A new nasopharyngeal swab RT- PCR was performed at 14 days of life and was positive for SARS-COV2. The Table 1 summarizes the results of the laboratory tests. Nosocomial infection was ruled out, a high nasal flow cannula was initiated, and oral azithromycin was administered at a dose of 20 mg/kg/day for 5 days. A nasopharyngeal swab analyzed via RT-PCR was negative at 21 days of life.

The follow-up showed mild respiratory worsening by day 35; a secondary infection was ruled out and the patient was switched to nasal continuous positive airway pressure. He was weaned to room air after 45 days of life and discharged by day 50. Follow-up visits were planned for one week and one month following the discharge.

## 3. Discussion

Mothers with COVID-19 infection may present with obstetrical complications such as preterm labor, premature rupture of membranes, intrauterine growth restriction and neonatal death. In our case, preterm labor is presumably linked to the mother’s COVID-19 infection, as there was no other evident cause. In neonates, COVID-19 infection in the mother may cause severe acute respiratory distress and biological disorders, including thrombocytopenia and abnormal liver function [4]. Laboratory tests in the present case showed a normal liver function and platelet count. Many cohort and case studies have strongly supported the absence of vertical transmission [5,6,7,8], However, Lan Dong et al. [9] reported a possible in utero infection in a newborn with elevated IgM antibodies to SARS-CoV-2 born to a mother with COVID-19. Alzamora et al. reported [10] a possible vertical transmission in a case report. Finally, the group from Antoine-Béclère Hospital (France) recently reported a proven in utero trans placental transmission of SARS-Cov2 from a pregnant woman infected with COVID-19 [11].

The mode of transmission in our case is most likely to be vertical given the short time it took for the neonate to become SARS-CoV-2 positive after delivery (24 h), the absence of skin-to-skin contact at birth and throughout admission, and the absence of visits by the mother as she was symptomatic. In addition, the baby remained in a closed incubator with droplet and contact precautions, as COVID-19 was suspected during delivery; therefore, it seems unlikely that the infant was infected by the nosocomial spread of aerosolized virus or by an infected healthcare worker. Breastfeeding was excluded as the route of transmission as the infant was exclusively formula fed. However, our report has some limitations, including the fact that it presents a single case, a vaginal delivery and due to the absence of serology testing and PCR testing of the amniotic fluid, placenta and umbilical cord, as well as the mother’s vaginal secretions.

Transmission through breastfeeding needs to be elucidated. In our unit, the use of maternal expressed breast milk is contraindicated in mothers with COVID-19. Skin-to-skin contact is not permitted given the benefit–risk balance. More studies are therefore needed in order to provide accurate evidence-based management guidelines. 

In the present paper, we noted a symptom-free interval between initial transient respiratory distress that was most likely related to the preterm birth, and the secondary respiratory distress that appeared by day 14, with a prolonged need for respiratory support and a longer length of stay compared to other reported pediatric studies [12,13,14].

This emphasizes the need for a cautious and close follow-up period for asymptomatic neonates born to mothers with COVID-19 infection who can develop severe pneumonia.

## 4. Conclusions

Premature infants born to mothers with a COVID-19 infection may also have a COVID-19 infection, presumably via vertical transmission. Further studies are needed to confirm this route of transmission. The need for an extended and cautious follow-up period for asymptomatic neonates with COVID-19 should be kept in mind, as symptoms may appear secondarily with rapidly severe respiratory symptoms that may require prolonged respiratory support.

## Figures and Tables

**Figure 1 children-07-00087-f001:**
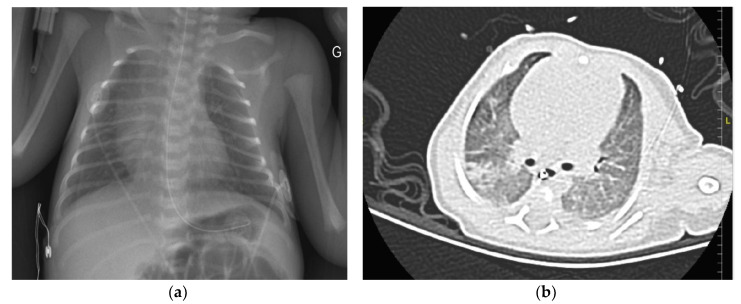
(**a**): Chest X-ray of a neonatal infant with transient respiratory distress on admission; (**b**): chest CT scan at day 14 of life, showing consolidation and ground glass opacities.

**Table 1 children-07-00087-t001:** Laboratory test results of the neonate.

	1 Day of Life	7 Days of Life	14 Days of Life
WBC (giga/L)	11.5 (5–20)	-	18.20 (5–20)
Lymphocytes (giga/L)	-	-	5.62 (2–11)
Platelets (giga/L)	125	-	151
CRP (mg/L)	<5	<5	<5
PCT (ng/L)	-	-	0.18 (0–0.5)
Ncov RNA throat swab	Positive	Positive	Positive
Creatine kinase (UI/L)	-	-	165
Aspartate amino transferase (UI/L)	-	-	48 (25–75)
Alanine aminotransferase (UI/L)	-	-	18 (13–45)
Blood cultures	Negative	Not done	Not done

C-reactive protein (CRP); procalcitonin (PCT); white blood cell count (WBC); novel coronavirus RNA (Ncov RNA).

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
