# Peer review of "Neonatal COVID-19 Pneumonia: Report of the First Case in a Preterm Neonate in Mayotte, an Overseas Department of France"

_children, 2020, doi:10.3390/children7080087_

Round 1

Reviewer 1 Report

The authors present a case of presumed vertical transmission of COVID 19 in a preterm infant. I believe that any new information regarding transmission and treatment of COVID 19 is essential right now. However, much has already been published about the peripartum period and COVID. This case is unique in the labor being preterm and the infant's course being particularly severe. I think that more discussion needs to be added regarding the nature of vertical transmission in this case and how it specifically adds to the literature.

The manuscript requires a great deal of English language revision. It should be fully proofread for grammar issues. I will not list errors here.

Abstract

Line 12 – I would refrain referring to “typical” imaging findings as COVID 19 can manifest in any way imaginable.

Introduction

Line 26 – population number seems to have a typo

Report

Line 1 – please provide background on “chronic bronchopneumonia.” This seems to be a vague condition to include in her medical history that has a lot of impact on how respiratory symptoms might be interpreted.

Line 39 – how long after delivery did the mother develop symptoms? Was she asymptomatic at last gestational visit?

Line 41 – when was the mother's COVID PCR checked? On admission or at development of symptoms?

Line 42 – describe radiographic findings. I would refrain from using the word “typical” as COVID can look like anything. How does current imaging compare to those obtained in past workup of “chronic bronchopneumonia?”

Line 53 – Was infant's COVID test a matter of routine due to mother’s positivity? Was obtaining serum IgM or IgG ab testing in the infant considered to better determine if infection occurred in utero? I am not familiar with treatment of COVID 19 in infants, are there any therapies available or under investigation?

Line 62 – GGOs appear bilateral to me, not unilateral, in Figure 3.1.b.

There is no figure legend for Table 1 (3.2). It is difficult to know whether this is data for the mother or the infant.

Line 68 – what was respiratory worsening due to? Was repeat PCR performed? The negative at day 21 could have been a false negative given persistent symptoms.

Discussion

Line 82 – please provide a better summary of what is known about vertical transmission of COVID 19. Does it occur in utero, does it occur during delivery or shortly thereafter? Anything known about C-section vs vaginal delivery? How does this case fit in to what is already known? When you suggest your case occurred due to vertical transmission, do you mean in utero or during delivery?

Line 89 – you state that COVID 19 infection was suspected during delivery. However, earlier you state that the mother developed symptoms shortly AFTER delivery and that was when testing was performed. Can you please explain this discrepancy. If COVID was NOT suspected, I think that it is entirely feasible that aerosolization of respiratory particles during a vaginal delivery might be a source of infection in the infant. Were healthcare workers in PPE? Did any of them develop symptoms?

Author Response

Dear Reviewer 

First of all, we would like to thank you for your instructive comments and suggestions that will help us to greatly improve the quality of our manuscript. We answered and corrected point by point the manuscript accordingly. Please find the comments and answers in the following table

Reviewer 2 Report

The authors report a possible case of true congenital infection with SARS-COV-2 in a preterm neonate.

The data provided is presented clearly.

However, there is incomplete laboratory evaluation to fully support the possibility of true congenital infection. It would be helpful for the authors to include the following information in the report:

  • Was the mother exposed to another known case ? What was the source of infection for the mother?
  • Is the potential mechanism of infection via. maternal viremia and antenatal infection just prior to delivery despite mother being asymptomatic, or exposure to vaginal secretions that could have been infected, although not tested?
  • How do you follow a positive test in the first 24 hrs of life in a newborn?
  • did you evaluate for other pathogens as cause of pneumonia in the newborn?
  • Was post-natal exposure/infection from a source other than the mother ruled out? Eg. other family  members of hospital staff?
  • Was antibody testing performed in the infant or the mother? If no, why? include this in the limitations section. If yes, please include results. 
  • It needs to be clear that without the information missing (as described in the limitations section) it is difficult to confirm congenital infection. Consider providing some guidance re. how these newborns should be evaluated.

Author Response

Dear reviewer

First of all, we would like to thank you for your instructive comments and suggestions that will help us to greatly improve the quality of our manuscript. We answered and corrected point by point the manuscript accordingly. Please find the comments and answers in the following table

Reviewer 3 Report

The authors describe the first case of vertical transmission of SARS-COV -2 in a preterm infant in Mayotte France.

To strengthen the case report kindle add the following:

Did this mother had prior preterm births?

Was the RT-PCR repeated between day 1 and day 14? Was this the same infection or a re-infection?

Were IgG/ IgM ever drawn on the infant?

Was placenta sent for pathology? What did it show?

What was the maximal respiratory support this infant ever needed? Did he require surfactant? How was the growth in the NICU?

Author Response

(The authors gave the same response as above.)
